# DATA-CENTRIC AI GOVERNANCE: ADDRESSING THE LIMITATIONS OF MODEL-FOCUSED POLICIES

## ABSTRACT

Current regulations on powerful AI capabilities are narrowly focused on "foundation" or "frontier" models. However, these terms are vague and inconsistently defined, leading to an unstable foundation for governance efforts. Critically, policy debates often fail to consider the data used with these models, despite the clear link between data and model performance. Even (relatively) "small" models that fall outside the typical definitions of foundation and frontier models can achieve equivalent outcomes when exposed to sufficiently specific datasets. In this work, we illustrate the importance of considering dataset size and content as essential factors in assessing the risks posed by models both today and in the future. More broadly, we emphasize the risk posed by over-regulating reactively and provide a path towards careful, quantitative evaluation of capabilities that can lead to a simplified regulatory environment.

## 1 THE SHORTCOMINGS OF TODAY'S AI GOVERNANCE

As AI has made its way to wider audiences, it has continued its rapid pace of development, giving everyday users highly specialized computing tools and capabilities. This raises questions for governments, academics, and commercial labs about whether certain AI capabilities or behaviors should be deemed as too "risky" for public access (Dragan et al., 2024).

Today's AI governance efforts have coalesced around the terms "frontier", "foundation", "dual-use", and "general purpose" to describe the largest, most capable of these models. In governance documents, models described by these terms are subject to additional scrutiny and regulatory interest. Despite general agreement for the types of AI-accelerated risks that regulations aim to curtail, there is less clarity and consensus on concrete definitions for such models. In an effort to define the characteristics of these large, capable models, a number of policy documents have focused on parameter counts and/or FLOPs, measures of model size and compute requirement (European Union, 2024).

We argue that this approach is short-sighted for three reasons. First, there is no consistent definition of "frontier", "foundation", "dual-use", and "general purpose" models. This lack of definitional clarity has led to a governance landscape with misguided quantities, such as FLOPs and parameters counts, and ceilings for what constitutes a covered capability; we elaborate on this in Section 2.1. Second, advances in efficient machine learning means that models require fewer parameters and FLOPs to achieve the same outcomes, resulting in capable models that fall below regulatory ceilings. Finally, the focus on the largest and most compute-intensive models ignores the fact existing, smaller models can be just as capable as their larger counterparts. These factors culminate in inadvertent loopholes that powerful capabilities can slip through, rendering expensive regulatory efforts not only useless but potentially detractory from beneficial uses of AI technologies.

The broader field of machine learning has recognized the role of data as a direct indicator of model performance (Hoffmann et al., 2022b; Ng et al., 2021), suggesting that dataset quality and size should also be included as factors in conversations surrounding model capabilities. In this paper, we first discuss the limitations of the current model-focused governance ecosystem. Then, we demonstrate the value of a data-focused approach to AI governance. In particular, we present experiments corroborating the role that dataset size plays in model capability. Finally, we propose legal and technical approaches to AI governance rooted in our understanding of the data-model relationship.

## 2 DEFINITIONAL CHALLENGES AND FLAWED LIMITS IN AI GOVERNANCE

Much of the conversation around AI regulation has centered itself around the prevention of behaviors that are deemed to be "harmful" or otherwise detrimental to society (Dafoe, 2018; Hoffman & Frase, 2023). The mention of "harm" is too often unqualified and does not address the capabilities of existing technologies that may already be capable of much of the malicious behavior discussed in AI policy circles today. For example, AI for biological agent design is widely cited as a potential harm (Callaway, 2024), yet computational drug discovery has been the norm since the 1980s and has enabled the discovery of drugs such as ritonavir, a medication critical in treating both HIV and COVID-19 (Van Drie, 2007). The conversation surrounding the use of AI to further societal harms must contextualize the additional marginal risk posed by these methods when compared to existing technologies such as search engines or statistical inference algorithms.

The AI governance ecosystem's difficulty in defining and identifying harm extends into fragmented efforts to define modern machine learning capabilities and the factors that make them powerful. In the following sections, we demonstrate the shortcomings and inconsistencies of these model-focused AI governance efforts, while identifying key drivers of AI risk that are currently overlooked in modern AI policy.

### 2.1 AN UNSTABLE DEFINITION FOUNDATION

The use of the terms "foundation", "frontier", "dual-use", and "general purpose" to describe machine learning models has arisen in the past few years in an effort to isolate classes of models seen as posing the greatest risk of harm to public safety. In 2021, Stanford University researchers introduced "foundation models" as a term of art in "On the Opportunities and Risks of Foundation Models" (Bommasani et al., 2022). The paper uses the term to describe machine learning models trained using self-supervised learning methods on large sets of data to the point that they demonstrate emergent behaviors during inference.

The term "foundation model" has spread swiftly throughout the AI research community to a point of saturation where any model trained on a subjectively large set of data can be termed "foundational." More recently, the terms "frontier model," introduced in "Frontier AI Regulation: Managing Emerging Risks to Public Safety" (Anderljung et al., 2023) and "dual-use model," found in the "Executive Order on Safe, Secure, and Trustworthy Development and Use of Artificial Intelligence" (The White House, 2023) and the EU AI Act (European Union, 2024), have arisen as similar descriptions of large, cutting-edge models with an increased potential for harm. The cross-cutting motivation of regulatory efforts has been that these types of models can pose serious risks to the general public and should be governed as such.

Different regulatory bodies have similar motivations for controlling AI models that they perceive as enabling risky behaviors. Despite shared goals, these efforts are not aligned with respect to the definitions they utilize to bound powerful AI capabilities. In Table 1, we highlight impactful papers and policies that have shaped international AI governance. In particular, we highlight the inconsistencies between how influential works which first introduced various terms and thresholds disagree from their actualization in policy proposals.

Terminology such as "foundation" and "frontier" are terms of art that have non-static and contentious definitions, suggesting that utility-based terminology such as "general purpose" may be better regulatory terms instead. Furthermore, a leading approach is to to bound "risky" AI models in terms of the amount of computation required to train them. As we demonstrate in Section 2.2, these thresholds do not appropriately bound "risky" AI models—a driving goal for regulatory efforts. Additionally, the documents that discuss training on "large" amounts of data do not define how many data points meet the bar, leaving leeway for bound parties to argue exemptions.[1]

### 2.2 CAPABILITY AND MODEL SIZE ARE NOT STRICTLY CORRELATED

Today's AI governance efforts regularly seek to define frontier models by their size and therefore by setting a regulatory threshold on the number of parameters included in a model. The rationale behind

---

[1]Historically, the computing paradigm of "big data" suffered from similar criticisms with no concrete amount or volume of data being defined for the purpose of strict regulation.

| | | Terms | SSL | Large data | FLOPs | Params. |
|---|---|---|:---:|:---:|:---:|:---:|
| **Terms** | Bommasani et al. (2022) | Foundation | ✓ | ✓ | – | – |
| | Anderljung et al. (2023) | Foundation, Frontier | ✓ | ✓ | $> 10^{26}$ | – |
| | Alstott (2023) | Frontier | – | – | $> 10^{26}$ | – |
| **Governance** | The White House (2023) | Foundation, "Dual-Use"[1] | ✓ | ✓ | $> 10^{26}$ | $> 10B$ |
| | Romney et al. (2024) | Frontier, General Purpose | – | – | $> 10^{26}$ | – |
| | European Union (2024) | General Purpose | ✓ | ✓ | $> 10^{25}$ | $> 1B$ |
| | Wiener et al. (2024) | Frontier | – | – | $> 10^{25} / 10^{26}$ | – |

Table 1: Variance in model definitions across policy documents.

this approach is a set of experiments that demonstrate that models with larger numbers of parameters, with all other factors held constant, suddenly perform drastically better on downstream tasks they are not explicitly trained for (Wei et al., 2022). This phenomenon was termed "emergence" and drove fears that sufficiently large models can perform well on tasks that pose risks to public safety.

Discussions prioritizing model size as a viable threshold are fixating on a superficial, easy-to-obtain quantity that is ultimately a red herring. In reality, model capacity and generalizability are characteristics that are innately difficult to quantify and measure. Not only are current generalization benchmarks lacking in accurate definitions for model capabilities (Raji et al., 2021; Ge et al., 2023), but it is common for smaller, more task-focused models to perform better than large, broad-purpose models on specific downstream tasks, as demonstrated below.

We use the task of image segmentation as an example where smaller models can outperform their larger counterparts. Specifically, we examine RefCOCO (Kazemzadeh et al., 2014), a common image segmentation dataset used to train vision-language models (VLMs), and two models which attain near-state-of-the-art performance on it, PaliGemma (Beyer et al., 2024) and UniLSeg (Liu et al., 2023). PaliGemma is a large VLM consisting of $3.0 \times 10^9$ parameters (Google, 2024). On the other hand, UniLSeg consists of only $1.7 \times 10^8$ parameters—an order of magnitude smaller than PaliGemma. Yet, UniLSeg achieves a mean intersection-over-union of 81.7 versus PaliGemma's 73.4 on RefCOCO, which is a massive gain of $\sim$11.3% in performance.[3] Figure 1 additionally demonstrates the performance of two more near-state-of-the-art models, UNINEXT (Yan et al., 2023) and HIPIE (Wang et al., 2023), on RefCOCO for completeness.

To further underline this point, we visualize the accuracy of top open-source language models on the Massive Multitask Language Understanding benchmark (Hendrycks et al., 2020) as a function of model parameter count in Figure 1. Low parameter counts do not imply incapability, demonstrating again that parameter counts alone are an insufficient quantity to define capability frontiers. More parameters are helpful insofar they can fit an appropriately larger amount of data—the two concepts must be bundled to properly circumscribe AI capabilities.

## 2.3 A MISPLACED FOCUS ON FLOPS

Definitions of foundation and frontier models (see Table 1) include regulatory thresholds defined by cumulative training FLOPs. Much analysis on the issue of FLOPs as a regulatory threshold was conducted by Hooker (2024). We extend this analysis in the following section and show that established FLOPs thresholds have no basis in outcomes or technical reality.

---

[2] Despite using the words "dual-use", the definition provided in the document are more aligned with accepted definitions of "general purpose."

[3] Model performance numbers are obtained from their respective papers and Papers With Code. Parameter counts are derived from the respective papers.

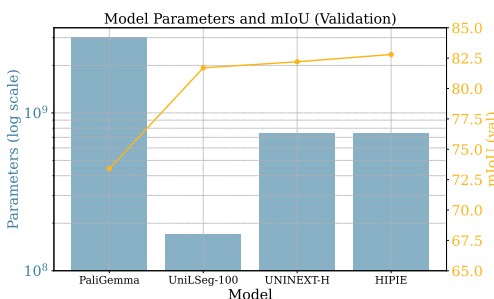 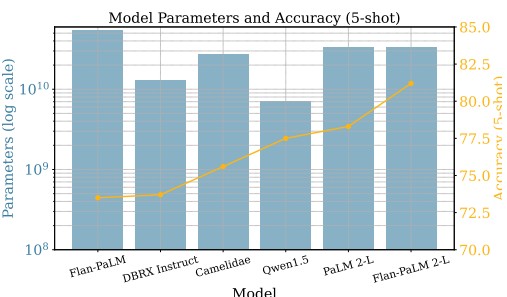

Figure 1: **The effectiveness of a model isn't solely determined by its size or computational complexity.** (Left) Despite PaliGemma having an order of magnitude more parameters than UniLSeg, it performs 9.4 mIoU points worse on the common RefCOCO (val) benchmark. (Right) Larger models do not necessarily perform better than smaller ones on the common MMLU benchmark.

Some of the largest models in existence today are sufficient to employ in harmful activities (OpenAI, 2024b;a), yet all fail to meet American FLOPs thresholds (see Table 2), raising questions about the threshold's usefulness. These same models are covered under the the EU's proposed threshold of $10^{25}$ for AI models. However, a fractured environment in which a model regulated in France might not be subject to the same regulations in the United States will lead to confusion.

These thresholds further exacerbate the perception that frontier capabilities can only arise from large models trained with a large amount of computation on larger datasets. As we further demonstrate in this section, even smaller models trained with fewer resources on smaller datasets can set a capability frontier. In fact, research incentives necessitate the creation of methods that reduce computational needs for model training—a trend that is contrary to regulatory assumptions.

**Optimizations reverse trends.** One way to visualize the futility of FLOPs thresholds is via recent works such as those on efficient sparse training (Chen et al., 2021) (Figure 2 (left)) or other architectural improvements (Zhu et al., 2024). They demonstrate that model performance can, in some cases, be decoupled from computational cost—models can train faster and more accurately with fewer parameters and FLOPs. Further research demonstrates decoupling in the opposite direction, i.e., efficient training can occur in compute-constrained environments. Models distributed across multiple machines can be trained with a fraction of parameters while equaling performance at the cost of increased FLOPs (Huh et al., 2024). In summary, policies solely relying on FLOP ceilings to bound "frontier" models are relying on simplified computing proxies that may not correlate to desired outcomes of controlling the spread of "risky" models.

Public disclosure of metrics such as FLOPs is beneficial, however, most well-known commercial AI models do not publicly disclose the amount of FLOPs utilized in the course of training their models. Open-source models, by definition, have exact FLOPs counts available. Below, we provide estimates of FLOPs for a variety of large vision and language models, both commercial and open-source. For proprietary models, these estimates are based on assessments from third-parties rather than concrete disclosures from the respective AI companies.

**Efficient methods develop rapidly.** AI research progresses rapidly and the development of efficient methods is an entire subfield with deep financial incentives. The amount of FLOPs needed for a given model architecture to reach a target performance threshold generally tends to drop significantly over a short period of time as the machine learning community identifies software and hardware optimizations for widely-used models.

To illustrate this point concretely, we consider various vision transformers[6] trained on the ImageNet-1K classification benchmark (Russakovsky et al., 2014). In less than a year, the ML research community increased the achieved top-1 accuracy on the benchmark from 81.8% to 84.4% while reducing the required FLOPs by 42% from 17.6 to 10.2 GFLOPs (see Figure 2 (right)). This trend holds true for large language models as well (Dao & Gu, 2024).

---

[6]DeiT, PVTv2, CaiT, CoAtNet, XCiT, Swin, MViTv1, MViTv2. Numbers are gathered from the MViTv2 paper and are on models using a comparable amount of computation.

| Model | Model Type | Estimated FLOPs |
|---|---|---|
| LWM (Liu et al., 2024) | Open-source vision model | $5.6 \times 10^{22}$ [4] |
| Gemma-7B (Gemma Team, 2024) | Open-source LLM | $2.5 \times 10^{23}$ (Ruan et al., 2024) |
| Qwen-72B (Bai et al., 2023a) | Open-source LLM | $1.3 \times 10^{24}$ (Rahman et al., 2024) |
| Falcon-180B (Almazrouei et al., 2023) | Open-source LLM | $3.8 \times 10^{24}$ (Rahman et al., 2024) |
| Claude-2 | Proprietary LLM | $3.9 \times 10^{24}$ (Rahman et al., 2024) |
| Llama-3-70B | Open-source LLM | $6.3 \times 10^{24}$ (Rahman et al., 2024) |
| ChatGPT-4 | Proprietary LLM | $2.2 \times 10^{25}$ (McGuinness, 2023) |
| Gemini 1.5 (Gemini Team, 2024) | Proprietary LLM | $5.0 \times 10^{25}$ (Rahman et al., 2024) |
| LVM-3B (Bai et al., 2023b) | Open-source vision model | $7.6 \times 10^{21}$ [5] |

Table 2: Large commercial and open-source AI models and their estimated FLOPs.

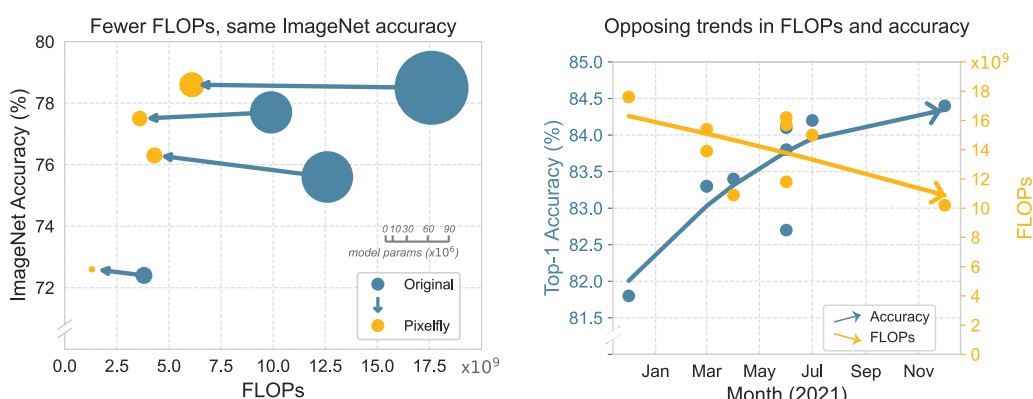

Figure 2: **FLOPs are insufficient determinants of capability.** (left) Pixelfly, a recent advancement in efficient model training, can maintain performance on ImageNet across many types of models while reducing their parameter counts and training FLOPs 68% and 200% on average, respectively. Each pair of dots represents a Mixer-S/B and ViT-S/B model and its Pixelfly variant. (right) The pace of FLOP count reduction is rapid as leading methods on the ImageNet benchmark drop FLOPs be 42% in one year while increasing accuracy.

**Test-time compute replace train-time compute for better performance.** Jones (2021) showed that a 10x increase in train-time compute eliminates about 15x test-time compute. Despite this, AlphaGo (Silver et al., 2016), Pluribus (Brown & Sandholm, 2019), and OpenAI's o1 (OpenAI, 2024c) all achieved drastically better performance over their respective baselines via test-time, compute-intensive search strategies. If this trend continues, then models trained for shorter amounts of time can achieve much better performance than their computationally expensive counterparts through the introduction of test-time computation strategies.

## 3 DATA IS MISSING FROM THE CONVERSATION

Machine learning capabilities are not singularly determined by their model architecture. Rather, machine learning capabilities are defined by *both* the model and the data provided. We define "data" as any information a model is exposed to, whether it is during training or deployment. This paper aims to center data in AI governance conversations. We suggest that models alone are not harmful; rather, the unique combination of models exposed to specific datasets (whether during training or inference) *and* subsequently being used for specific purposes may pose a risk to public safety (Baldridge et al., 2024).

Traditionally, training data (both pre-training and fine-tuning) was the only source of information that a model would have access to before making a prediction. However, models can now incorporate new, unseen data during inference through frameworks such as prompting and Retrieval-Augmented Generation (RAG) (Lewis et al., 2020). Therefore, *both* the training and deployment data are relevant when considering how a model incorporates information in its outputs.

## 3.1 BIG DATA TO USABLE INFORMATION

The rapid rise of AI since approximately 2010 can largely be attributed to (1) advancements in computational hardware in accordance with Moore's Law, and (2) a focus on large quantities of data. Models are useless without data, and the availability of "foundational" datasets, such as ImageNet (Deng et al., 2009) and Common Crawl,[7] brought modern machine learning capabilities to bear. AI datasets today, created through Internet scraping, are often orders of magnitude larger.

Dataset size is a key component in "scaling laws," or predictions of performance within a family of models as a function of variables in a training recipe. Research in this area finds strong relationships between model performance and amount of training data, amount of computation, and model parameters (Kaplan et al., 2020; Hoffmann et al., 2022a; Zhai et al., 2022; Google, 2023). Additionally, both Hoffmann et al. (2022a) and Google (2023) find that model and optimal dataset size scale at *equal proportions* as training compute increases.

However, even an optimal training recipe with an appropriate amount of data, parameters, and compute does not necessarily produce a useful model. The dataset *content* is a crucial factor. A model "trained on the internet" can unsurprisingly exhibit the same bias (Fleisig et al., 2024) and toxicity (Liang et al., 2023) present in the data and also fall short in other areas: it may fail at logical reasoning (Berglund et al., 2023), algebraic computation, or following a user's instructions, to name a few examples. In a limiting argument, a multi-trillion parameter model trained only on Shakespeare novels may never be able to reason about chemical weapon design.

To address this, models are fine-tuned on higher-quality, curated data. Popular techniques that rely on high-quality data include instruction tuning (e.g., reinforcement learning from human feedback, or RLHF (Ouyang et al., 2022)), training models to use tools or act as agents (Schick et al., 2023), or supervised fine-tuning for a specific task, such as generating images in a particular artistic style.

There is evidence that with the right data and training regime, models in the millions or single digit billions of parameters can perform comparably, if not better, than counterparts orders of magnitude larger in many domains (Yu et al., 2023; Yuan et al., 2024; Eldan & Li, 2023). In fact, once a model is sufficiently large, focusing on improving the quality and utilization of data can yield greater gains in performance for a task over simply increasing the size of the model. For example, the Retrieval Augmented Fine-Tuning (Zhang et al., 2024) framework has been shown to improve the question-answering performance of a 7B parameter Llama2 language model over that of GPT-3.5, which otherwise significantly outperforms it out-of-the-box.

## 4 DATA-CENTRISM OPENS NEW ANALYTIC FRONTIERS

Modern machine learning methods are useful beyond traditional data querying and correlation tools such as search engines in part due to their ability to retrieve, compile, and organize data even when given unspecific queries. Below we outline two distinct features that are uniquely enabled by the combination of ML models and data: (1) *retrieval*, where a model outputs information retrieved directly from its data, and (2) *derivation*, where a model compiles or synthesizes items from provided data to generate new information. These features enable new ways to interact with complex data that would otherwise be difficult to manage, offering potential benefits as well as risks, which we explore further below.

### 4.1 RETRIEVAL

As our ability to collect and maintain digital information has soared over the last few decades, retrieving the right results for a certain query has become a core technological focus. Billions of

---

[7]https://commoncrawl.org/the-data/

dollars have been spent towards developing efficient data representations for search engines (Brin & Page, 1998; Dean & Ghemawat, 2008) and databases (Corbett et al., 2012; Shvachko et al., 2010), and towards creating the algorithms to find and retrieve these results. Now, AI models trained on large amounts of data have become both capable encoders and retrievers of data (in addition to generators, as we describe in the section on derivation). This becomes a problem when a model has been exposed to specific data points that would be considered sensitive if directly retrieved, such as credit card numbers or classified information. The retrieval itself could occur either through (1) a model memorizing and then reproducing points in training data, or (2) retrieving from a large amount of data provided at test time, such as a company's internal database. Below, we describe these two cases in more detail.

**Retrieval from training data.**  Datasets contaminated with outliers have historically relied on dataset volume to dilute outlier effects. This leads to the misconception that a small quantity of "harmful" data points can be negated by massive amounts of otherwise commonplace data. Unfortunately, this intuition does not translate to modern machine learning methods. Large models are known to memorize some parts of training data and reproduce them if queried correctly (Carlini et al., 2022). Therefore, large models can retrieve, and therefore utilize, harmful data even if it is present in a negligible quantity.

However, memorization does not occur across data equally: prior work shows that "average" training samples are less likely to be memorized, whereas outlierse and duplicated data points are more likely to be memorized (Feldman & Zhang, 2020; Feldman, 2020; Carlini et al., 2022). As certain types of data, such as child sexual abuse material, are outliers on the Internet (Thiel, 2023), memorization of such data poses an inherent risk in the downstream usage of affected models, especially combined with the powerful retrieval abilities of current models. Nasr et al. (2023) is another example of work where ChatGPT was used to retrieve training data which comprised personally identifiable information of dozens of individuals.

**Retrieval from previously unseen data.**  An AI system may also be exposed to entire new domains of data during inference that were not present during training. Models can utilize new, unseen data through prompting or integration with external databases. Models' ability to effectively interpret new domains without prior training marks a significant shift in how we store and use information. Instead of creating expensive, specialized systems to process data like financial documents or hospital records, modern general-purpose models can understand and work with novel data formats they have never encountered before while requiring minimal engineering effort.

Many of today's large models are being specifically designed to respond flexibly to new tasks and prompt formats. In-context learning (Brown et al., 2020) allows users to provide example input-output pairs of a task to a large model which can equip it to solve novel instances of that task. Further, modern AI systems may be used to efficiently sift through large amounts of data at inference time—even if they have not seen it before—using frameworks such RAG (Gao et al., 2024). In RAG, given a user query, an answer can be generated by efficiently searching a database for relevant concepts and making sense of this new information to return a relevant response (Yasunaga et al., 2023; Kong et al., 2024; Blattmann et al., 2022). As these systems can now return sensitive examples not seen before by dynamically augmenting their knowledge or understanding new tasks from test-time examples, they can therefore be used in the furtherance of actions that pose risks to society by drastically lowering the boundary to both finding and exploiting risk-posing information (Barrett et al., 2023).

### 4.2 DERIVATION

As AI capabilities increase, a growing concern is the generation of original or derivative information that is more revealing than the data provided to the model. For example, if a system is given two entry-level textbooks in physics and chemistry, respectively, and uses independent concepts from either to build a toy rocket, we term the process of arriving at the toy rocket instructions "derivation."

This feature is especially present in modern machine learning methods when compared to technologies such as databases due to their ability to synthesize unrelated pieces of information on the fly. While retrieved content is often straightforward to recognize and check—i.e., it may be quickly obvious that a generated phone number is real, and possible to check if a particular image was con-

tained within training or deployment-time data—derived content is more nuanced and difficult to measure, and thus may present a greater concern.

Under this category, multiple pieces of otherwise mundane information could be compiled to form information that is now sensitive. For instance, a language model trained for code generation could be provided a description of a vulnerability and be used to generate code for exploiting it. Models have already begun to present synthesis capabilities in different arenas, such as for code generation of programming languages with low data availability (Mora et al., 2024) and the generation of Mathematics Olympiad-level geometric proofs as part of larger pipelines (Trinh et al., 2024).

The maximal extent to which current models are capable of derivation is not yet clear as methodologies for inducing such capabilities are constantly evolving. For example, although modern language models have shown nascent indicators of capability to generate novel research ideas in fields such as natural language processing, the ideas they generate lack diversity and may not be tractable (Si et al., 2024). Modern image generation models struggle to synthesize images precisely adhering to descriptions of unique combinations of objects and their attributes previously unseen in training data (Huang et al., 2023). Our intent in this section is not to establish a measure for models' derivation capability but rather to bring attention to derivation as a unique capability offered by modern ML models.

## 5 Avenues for Data-Forward Regulation

Given our analysis above, the inclusion of data in nascent AI governance conversations can simplify the regulatory overhead by enabling the use of existing legal frameworks and the creation and execution of novel, data-backed evaluation schemes. Specifically, there are numerous policies and laws surrounding the appropriate use of data in contexts that are deemed to be of risk to the public. Instead of reinventing these policies using a new set of definitions that are model-specific, expanding and modifying them to account for the use of data by powerful models might offer a simpler path towards effective evaluation frameworks in areas where definitions alone are vague, leading to simpler regulations.

### 5.1 Applying Existing Data-Focused Legal and Regulatory Approaches

Significant work has been and continues to be done to mitigate malicious model outputs or behaviors. Thus far, model creators have relied on identifying malicious outputs or behaviors through red teaming and safety training (Ganguli et al., 2022; Wei et al., 2023).

However, some classes of outputs or behaviors that are deemed risky could more easily be stemmed by careful curation of datasets. Unique information such as the relationship between a person and their social security number, or specific instances of child sexual abuse material, is extremely unlikely to be generated if that data is never provided to a model.

There exists a range of legal and regulatory frameworks that cover many categories of model outputs that are of greatest concern, including personal identifiable information, child sexual abuse material, and classified content. Data-centrism prevents models from acquiring the capacity for harmful behaviors prior to the expenditure of computation. Since existing regulations can be applied, AI governance can be achieved without the need for new regulatory frameworks.

### 5.2 Technological Levers for Data-Forward Regulation

Although research has shown that certain model capabilities emerge once sufficient model size and compute are attained (Wei et al., 2022), establishing regulatory thresholds is ill-defined given just these two metrics. As discussed in Section 3, models provided with the right data can perform comparably to, if not better than, larger and more compute intensive alternatives. Further, a model must first be paired with sensitive information for it to make use of it. That is, the model does not exist in a vacuum, and a data-forward approach that prioritizes data content and quality filtration over model size and computation could yield greater benefits in mitigating risks posed by the use of models. Here, we briefly outline examples of existing techniques and argue for the development of new methods.

**Existing data filtration.** Modern web-scale datasets are extremely large, numbering in the billions to trillions of data points. As such, human review of every data point is not possible from either a labor or monetary perspective. However, the volume of data does not permit the abdication of responsibility or duty to curate datasets responsibly. In response, methods have been proposed to partially or fully automate the filtration process (Albalak et al., 2024). Content can be filtered based on fixed patterns such as blacklisted source URLs or key words, however, these methods can be rigid and insensitive to the nuance of usage context. Large vision and language models such as CLIP (Schramowski et al., 2022) and Meta's Llama Guard (Inan et al., 2023) have been used to classify whether data points are risky under human-defined criteria and can be more sensitive to context than blacklist-based methods. However, these methods are far from perfect—offering an important avenue for future research.

**Quantifying risk for workloads.** In addition to data filtration schemes, a rigorous evaluation framework for powerful AI models that is inclusive of both models and data is needed. Many approaches are feasible, and we detail an evaluation framework under development that attempts to solidify this discussion into a quantifiable benchmark.

For example, imagine asking a model a question in a setting where accuracy of the answer matters, say "what materials make up Saturn's rings?" Short, broken answers such as "rock, water" would be regarded as unreliable or incorrect as opposed to an answer that demonstrates mastery of grammar and facts such as "The rings of Saturn are primarily composed of countless small particles of ice and rock. These particles range in size from tiny grains of dust to larger chunks that can be several meters across."

For a specific type of output, there is likely a minimum size threshold for a model to be capable of learning the syntax of that output domain (Chen et al., 2024). The initial stage of model training is focused on acquiring *fluency*—object detection models learn what the shape and proportion of a valid detection looks like, and language models learn the underlying structure and grammar of the languages over which they operate. In this stage, models are parameter-bound—the largest gains in fluency are likely to come from making models bigger. However, once a model has passed this hypothesized stage to learn the syntax "well enough," we posit that the model is now data-bound and improvements to performance, or correctness, are more likely to come from improvements to the content and utilization of data rather than just from arbitrary scaling (Wei et al., 2022; Yu et al., 2023; Eldan & Li, 2023).

This inherent relationship between fluency and correctness can be used as a powerful tool to regulate AI capabilities in a data-parameter inclusive fashion. For any arbitrary task, the performance of a model on that task can be plotted on a fluency-correctness curve. Once all workloads are plotted, the resulting risk profile can be adjudicated and a resulting judgment—whether a reduction in the parameter count of the model or a specific pruning of the training dataset is necessary—can be made by the model developers.

Ultimately, such an evaluation framework can aid in the development of regulatory system through which the government and model developers can safely, privately, and precisely iterate on removing the ability of models to aid in risky tasks prior to model release.

## 5.3 INCENTIVIZING DATA GOVERNANCE TOOLS AND PRACTICES

Just as existing policies and regulations advocate for the standardization of model documentation, such as model or system cards (Mitchell et al., 2019), data-centrism motivates the standardization of dataset documentation. Comprehensive approaches for doing so have been proposed already, such as Datasheets for Datasets (Gebru et al., 2021) or Data Cards (Pushkarna et al., 2022). These documentation formalisms currently detail dataset properties regarding content, structure, preprocessing, distribution, and intended or potential use cases. Given the common practice of aggregating datasets from multiple sources, mechanisms for documenting and tracking the provenance of dataset contents, such as Data Provenance Cards (Longpre et al., 2023), would greatly ease verification of information available to a model. Further, standardized ontologies (Zeng et al., 2024) that categorize and rank "risky information" can be applied to each dataset in a provenance card, which can then be used as a first approximation of potential retrieval and derivation capabilities.

In practice, red teaming (Perez et al., 2022; Bai et al., 2022) has become the standard to evaluate whether models intended for release pose a risk to public safety. However, red teaming is, as of yet, not standardized. Further, with the rapid increase in the amount of models that need to be assessed, there exists no mechanism through which the potential of models to perform specific tasks can be estimated before training them. The development of a technical framework for measuring the dynamics of the performance of a model family for a given task as a function of both model scale and quality of training data, particularly one that can identify inflection points at which a model's performance becomes *bound by its data* rather than its size, would be an important tool for more precisely identifying when models could feasibly be used in the furtherance of behaviors that harm society.

# 6 CONCLUSION: EVOLVING AI GOVERNANCE ALONGSIDE AI TECHNOLOGY

Despite rapid growths in both model and dataset sizes in recent years, AI policies have hinged on thresholds, definitional concepts, and qualifiers that limit their medium-to-long term liability. For a technology that will be with us for the foreseeable future, we can, and should, approach governance in a more deliberate manner, with a clear understanding of what enables these capabilities to be powerful in the first place.

Similar to how an arbitrarily large engine, no matter how specifically quantified, would be useless without defining the kind of fuel used with it, the AI policy landscape mistakenly focuses on a small set of model-based thresholds, particularly FLOP and parameter counts. Neither fully define how powerful a machine learning model may be without an understanding of the data that accompanies them. Furthermore, the lack of definitional clarity with what constitutes a "frontier", "foundation", "dual-use", or "general purpose" model complicates governance efforts. More generally, these two trends in governance further propagate the outdated idea that the largest, most compute intensive models are those which drive AI risk. As we reach a point where smaller models, when paired with large, foundational datasets or small, high-quality datasets, can perform as well as larger models, this narrow approach creates loopholes and unfairly penalizes otherwise beneficial technologies.

Centering data offers a more durable approach to AI governance, particularly as trends in quantifiable measures of model capability are difficult to predict. A focus on data also provides an opportunity to better research, define, and respond to benefits and risks posed by AI, a debate that remains nebulous in both policy and technical circles. Centering data also provides avenues for existing regulations surrounding sensitive types of data to apply while also clearing the way for new evaluation methods to quantify the use of data and models together. Expanding model-based regulations to focus additionally on their paired data builds a stronger foundation that is less prone to collapse.

While a pivot in the governance landscape may be daunting, a focus on data provides the opportunities and incentives for government, academic researchers, civil society, and the private sector to develop new tools and approaches that lead to meaningful policies.

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

## A Assumptions and Limitations

Given the rapid pace of AI development, we acknowledge the limits of our core analytic assumptions, grounded in the current state-of-art in the field, that drive the analysis and recommendations in this work. If these building blocks are outpaced by future developments, then this work should be revisited.

**Assumption 1: Powerful models are unable to reason without memorizing information.** Large models can perform well by both learning generalizable semantics over their training data, but also through the rote memorization of data or concepts. Currently, there exists no class of powerful machine learning models which are able to "reason" about the world without having memorized any data during its training period. Put another way, there are no reasoning agents that are derived in a manner that is completely detached from data. One can argue that such a model, should it exist, would fit the definition of "artificial general intelligence" as it could generalize to any new set of data without inherent data priors.

**Assumption 2: Dataset distillation methods are still over the horizon.** The field's understanding of the amount of data points needed for a model to achieve proficiency on specific tasks is still evolving. This area of research is termed "dataset distillation" and aims to reduce the number of data points necessary to achieve target metrics (Wang et al., 2020; Zhao et al., 2021). Further, it remains unclear what exactly constitutes a "data point," especially with modern methods like transformers, which rely on tokens, the amount of which varies with different tokenization methods (Sennrich et al., 2016; Schuster & Nakajima, 2012). We aim to establish one rigorous definition of "data point" in future work, as well as analysis of how many data points define emergent capability.

In a limiting argument, should data distillation methods improve to the point where models can learn generalizable knowledge without any data at all, this work would need to be revisited.

