# OpenReview forum: "Data-Centric AI Governance: Addressing the Limitations of Model-Focused Policies"
_ICLR.cc/2025/Conference — Submitted to ICLR 2025_

### Official Review · Reviewer_ELVC · 2024-11-01

**Soundness:** 3
**Presentation:** 2
**Contribution:** 4
**Rating:** 6
**Confidence:** 4

**Summary:**

Summary: This paper is essentially a position and framework paper arguing for new criteria for applying model governance policies (i.e., rules about model deployment, use, etc.). The paper critiques FLOP-oriented frameworks for measuring model size (and more generally highlights a major concern that ongoing AI Governance debates are not even using a shared language). The argumentation in the paper roughly flows as such: current “foundation/frontier” focused definitions are not unified enough, compute is not the primary determinant of capabilities, data is under discussed, and thus future policy discussions should reorient around definitions/taxonomies/criteria-to-regulate that include data.

The paper has a compelling point. The literature that’s engaged with here is reasonably well-selected (of course, given the volume of publications in the LLM and AI Safety space, being opinionated is required here and some readers may be upset about certain omissions). I expect the general policy directions illustrated here to be impactful.

However, the main issue in my view with the current draft is mainly the room for organizational improvement. Revisions that focus on hand-holding the reader about what the contribution is (e.g., should I be turning the page to find empirical experiments with data scaling) and summarizing the core argument could greatly strengthen the impact of this paper. I believe these revisions can be made within a relatively short cycle and am hopeful about this paper. I look forward to reading the next draft. Below, I summarize key feedback for this paper:

**Strengths:**

Strengths
* The core argument is strong here – I think the paper will successfully convince readers to adjust some of their future behavior (more research/policy along the lines outlined here, more efforts to unify language in policy debates).
* There is a reasonable selection of literature that is discussed. While there is certainly room to discuss more avenues of “data-centric” ML (esp. e.g. economics of data, data as an arena for new kinds of negotiations -- specific examples in the next section), I think the paper already has a very broad scope.
* Building off above, to be clear, I think future versions of this paper can have very high potential impact (i.e., I was convinced that indeed the policy directions here would be very beneficial if adopted).
* If this paper is able to help "clean up" terms in the AI governance community, this will go a long way.

**Weaknesses:**

Weaknesses
* In the current draft, it is not immediately clear what the kind of contribution is. Is this a framework paper, a position paper, and empirical paper, etc.? By the end of the paper, readers will know, but this could be communicated much earlier on.
  * To be specific, this paper could benefit from a very signpost-style paragraph, sentence, or bullet points in the last paragraph of Introduction that clarifies what new data and experiments are being presented here vs. what new argumentation is being presented.
* More generally, the organization of the paper (especially around the middle) could be improved, even with just some general signposting and clarification of how much of the arguments here are being summarized for the first time vs. presented for the first time. To be specific on this front, the current Section headers do give a nice sense of the topical coverage of the paper but do not clarify what kind of information will be presented in this section (argument vs. experiment vs. synthesis vs ...).
* Venue fit: I could see some readers expecting more of a traditional methods/results organization for the paper. Perhaps some readers would be more impacted by a traditional experiment-focused paper. Personally, I think the best version of this paper need not stick to a very traditional organization structure, but I just wanted to flag that this could be one direction of revision to think about.

**Questions:**

My main question that arises draws on the above areas for improvement: what is the primary contribution of this paper and what is the key narrative you’d like to see researchers take away vs. what should policymakers take away?

Building off the above references (influence, data deals, data as leverage), to what extent does future progress on training data influence measurement techniques, new practices around "data deals", new practices around data-related collective action (e.g., action by artists or writers) affect the framework you've constructed here?

If successful, can a data-based framework replace most discussion of compute governance? Should we be working towards some kind of hybrid?

---

> ### Author Response · Authors · 2024-11-19
>
> Thank you for your thoughtful and detailed review. We are delighted that you found our work compelling and impactful. Your insights have helped us refine how we present the paper’s contributions and organization, ensuring that its outcomes are clear and accessible.
>
> To summarize the central motivation of our work: Current AI governance is disproportionately focused on the “engine” (compute) while largely overlooking the “fuel” (data), despite the fact that the former is ineffective without the latter. We provide a pathway towards data-centric AI governance by analyzing the unique qualities of AI combined with data and discuss the need for evaluation frameworks and benchmarks that offer effective and flexible levers for future regulations.
>
> **What kind of paper is this?**
>
> We recognize the need for clearer communication about the nature and contributions of this paper. To address this, we will update the abstract and the final paragraph of the introduction to explicitly summarize the paper’s key contributions and intended outcomes. This is an AI policy paper that aims to engage the machine learning community in discussions about data-centric governance and inspire the development of new evaluation frameworks and benchmarks. These updates will ensure that the paper’s contributions are evident to readers early on.
>
> **Venue fit**
>
> We carefully considered whether this paper would be better suited for a policy-focused venue. Ultimately, we concluded that ICLR is the most appropriate forum because the challenges we address require the technical expertise and active engagement of the machine learning community. Policy-focused venues often lack the audience capable of implementing the proposed evaluation frameworks and benchmarks. Moreover, future regulatory frameworks will likely draw on input from AI researchers, making it crucial to introduce these discussions at venues like ICLR.
>
> While some readers may expect a traditional methods/results structure, we believe that ICLR is an appropriate venue for qualitative research and policy discussions, particularly when these discussions are deeply intertwined with technical advancements in machine learning. Our paper is intended to circumscribe the problem and highlight the importance of a data-centric approach to AI governance, paving the way for future work that will focus on specific technical contributions.
>
> **Addressing broader questions**
>
> We also appreciate the broader questions you raised about the implications of data-centric governance. For example, you highlight practices such as “data deals” or collective data-related actions (e.g., by artists or writers) could significantly influence the framework we propose. While our current paper focuses on foundational principles, we see this as an important area for future exploration.
>
> Similarly, we agree that data-centric governance does not necessarily replace compute-focused governance but complements it. In fact, we envision a hybrid approach where compute- and data-centric metrics are used in tandem to provide a comprehensive understanding of model capabilities and risks.
>
> ---
>
> Once again, thank you for your constructive feedback and for highlighting areas where the paper can be further improved. We look forward to incorporating these revisions to strengthen the paper and ensure its contributions are clear and impactful.

---

> > ### Comment · Reviewer_ELVC · 2024-11-19
> > **Helpful response**
> >
> > Thanks to the authors for this response. This is helpful and improves my confidence that this paper can be turned into a very high impact contribution with a relatively minor set of contributions. Looking at other reviews, I also wanted to clarify my position that while some readers might be surprised with venue fit, I am not arguing against the fit in my review -- I think this paper may stand out, but agree that the core readership will likely be more on the technical side than policy side.

---

### Official Review · Reviewer_SLu2 · 2024-11-04

**Soundness:** 1
**Presentation:** 2
**Contribution:** 1
**Rating:** 3
**Confidence:** 5

**Summary:**

- This paper argues that current governance/regulatory debates pay insufficient attention to the role of data, instead focusing myopically on model parameter counts/FLOPs
- It provides three justifications for this argument:
   - First, there is no consistent definition of “frontier”, “foundation”, “dual-use”, and “general purpose” models.
  - Second, advances in efficient machine learning means that models require fewer parameters and FLOPs to achieve the same outcomes, resulting in capable models that fall below regulatory ceilings.
  - Finally, the focus on the largest and most compute-intensive models ignores the fact existing, smaller models can be just as capable as their larger counterparts.
- The paper then argues for a dataset-centric approach, and presents experiments highlighting the role dataset size plays in model capability

**Strengths:**

- The paper is clear in its writing.
- The paper touches on a debate of significance and widespread relevance.

**Weaknesses:**

- There’s no novel technical contribution (theory, technical framework, algorithms, dataset). This is a policy position paper, and as such, I think the better venue is a conference/journal with a policy focus.
- One complaint is that there’s ambiguity regarding certain terminology (e.g., “foundation models,” “frontier,” etc.). But isn’t resolving definitional ambiguity the role of legislatures and courts? Ambiguity is unavoidable–but most countries have enforcement or judicial processes for navigating and resolving it. Sure, the term "foundation model," might be ambiguous now. But if its used in a law, then courts (or agencies) will provide definitional clarity to it.
- It’s not clear how the paper extends prior critiques of FLOPs thresholds. The points made here seem to be ones made in prior debates.
- The paper suggests that policy debates should instead focus on the role of data. But it isn’t precise about what, specifically, this might entail. And it doesn’t address how governance anchoring in data should be designed. Are the authors suggesting the equivalent of FLOPs thresholds for data? I.e., thresholds based on dataset size? Nor does the paper address the difficulties that arise when designing regulation around data. For instance: how should different types of data be treated? What distinctions should be drawn between human and machine-generated data?

**Questions:**

See above!

---

> ### Author Response · Authors · 2024-11-19
>
> Thank you for your helpful comments! We appreciate that you find the paper well-written and topical. We address each question/concern below.
>
> **Relevance to ICLR audience**
>
> We wholeheartedly agree that this is an AI policy position paper, and will make that distinction more explicit in our introduction to ensure clarity.
>
> While we recognize that policy-focused journals may traditionally host such discussions, we firmly believe that ICLR provides a uniquely important venue for this work. The ICLR community—composed of leading machine learning researchers and practitioners—plays a pivotal role in advancing the discussions outlined in our paper. Many of the challenges we address, such as the development of new evaluation frameworks and metrics, depend directly on the engagement of the ICLR community to ensure their feasibility and adoption.
>
> Additionally, this paper aims to bridge the gap between governance discussions and the ML research community by conducting a meta-review of the field and proposing actionable directions rooted in technical expertise. By doing so, we hope to empower the ICLR community to actively contribute to shaping the policies that will ultimately guide the future of their own work. While a policy-focused journal may have been an acceptable venue for our work, we specifically want to engage those in a unique position to shape future policies in ways that are informed by technical expertise and grounded in the realities of ML research and development. We hope that the ML community will see this work as an invitation not only to advance the field technically by developing more efficient training methods or models with better scaling properties, but also to participate in cross-disciplinary dialogue and influence how their innovations are governed, used, and understood by policymakers and broader societal contexts.
>
> **Ongoing definitional challenges**
>
> We appreciate the point about judicial and legislative bodies eventually providing definitional clarity for terms such as “foundation models” and “frontier models.” However, it is precisely this potential misalignment between legal definitions and the understanding of practitioners that motivates our paper. As highlighted by examples from other fields—such as the California Supreme Court’s classification of bees as fish (Almond Alliance of California v. Fish and Game Commission)—definitions established through judicial or legislative processes may diverge significantly from the operational realities understood by domain experts.
>
> The debate over definitions like “foundation” and “frontier” is still active within the AI research community. By addressing these ambiguities now, at venues like ICLR, we can contribute to a shared understanding that is both technically rigorous and practically relevant. This proactive approach ensures that future legal and regulatory frameworks are informed by the expertise of those who build and study these systems, rather than relying solely on external processes to resolve issues that originate within the ML field.
>
> **Paper contribution to prior critiques of FLOPs thresholds**
>
> Thank you for bringing up that clearer contextualization is necessary for our work. We have addressed this point in the global response and added the proper clarifications in the paper.
>
> On the topic of data-centric governance, we acknowledge that our initial draft could have been more clear about what this approach might entail. To address this, as outlined in our response to Reviewer SyiS, we will clarify our proposal for assessing data-related risks through standardized ontologies of “harmful” data in Section 5.2.
>
> ---
>
> We hope that this clarification helps articulate how our paper seeks to advance the discussion on FLOPs thresholds and data-centric governance.

---

> > ### Comment · Reviewer_SLu2 · 2024-11-21
> >
> > Thank you for your comments!! I still maintain the concerns articulated in the original review, so I'll keep the score.

---

> > > ### Author Response · Authors · 2024-11-23
> > >
> > > Hello! We would love to discuss the paper more with you as your feedback will help make our paper better. Are your concerns about venue fit or about whether this is a judicial or legislative issue?
> > >
> > > As the discussion period does not end until the 26th, we'd appreciate the opportunity to improve the paper in the time that we have with you.
> > >
> > > Thank you,
> > > Authors

---

### Official Review · Reviewer_SyiS · 2024-11-05

**Soundness:** 2
**Presentation:** 4
**Contribution:** 2
**Rating:** 6
**Confidence:** 4

**Summary:**

This paper provides the motivation to move from regulating AI models based on characteristics such model size, number of FLOPS per training, or whether or not the model is classified as frontier/foundation. The authors instead motivate us to focus on data, and more important the content of the data and not the size of the data. The logic is, if harmful data is not in the dataset then the model cannot create harmful outputs. So, if people's private data is not in the dataset, then those people's privacy cannot be compromised.
The main claims are that frontier/base/foundation model are not well defined and the difference between each is unclear, and this makes legislating based on this classification possibly incosistent. Additionally, using FLOPs as a an indication of how powerful a model is, that too is inconsistent because smaller models which take less computation to train can be just as capable.

**Strengths:**

- Clarity: This is a very well written paper and the arguments are well laid out
- Significance: This is an important area of research for which there are no clear answers currently. Contribution to these discussion are very valuable.

**Weaknesses:**

Novelty: The importance of the role of data in developing safe and fair AI system is perhaps one of the most documented. Not only in foundation models but in previous deep learning applications.  The idea that in thinking of regulation the field is only thinking about the algorithms and not the data is not a correct assumption; in fact the works the authors cite, on Datasheets by Gebru et. al, those were motivated by the discussions on the importance of data and the contents of datasets in designing safe system. From the datasheets, we have seen Data Nutritional labels by Holland et. al and we have seen many more expansions of these concepts to  applications like Health (Rostamzadeh et. al. 2022)
One of the strongest points the authors make in their paper is the implications of derivation and summarisation over multiple documents and information sources which are skills unique to the current machine learning models. They mention how these abilities mean that information that is benign in isolation can now be aggregated into unsafe or undesirable model outputs. However, it is not clear what the authors would suggest as a solution to this problem because as they said, each piece of data, in its originating dataset was not harmful, it is the aggregation and derivation that creates harmful outcomes. The threats, especially to privacy posed by aggregation are known (see literature on anonymisation strategies) but perhaps here the authors bring these back up because of the scale at which these models can work. If this is the case, the authors should better support this. If not this, then the only other solution I can think of is that we imagine how our datasets can be combined with other data to create harmful outcomes  -- and I think this is an incredibly difficult ask.
Overall I failed to see the novelty of this work especially with the work from Hooker 2024, in which the arguments against the user of FLOPs are well laid out and keeping in mind all the important work that has gone into the field to convince us on the importance of auditing the suitability of datasets for specific projects.

**Questions:**

- In this new paradigm of more capable models, what additional characteristics of datasets should be considered for legislation?
- How do we propose people audit their datasets differently to make sure that they are building over safe dataset?
- What do we have to do differently when we build datasets for this new iteration of models?

---

> ### Author Response · Authors · 2024-11-19
>
> Thank you for your clear read of the paper and helpful suggestions about where we can improve the work. We agree that this is an important area of research and debate, especially amongst the community of researchers present at ICLR.
>
> **Prior work on FLOP thresholds**
>
> First, we acknowledge a significant oversight in failing to include Sara Hooker’s 2024 paper in our citations. This was unintentional, especially given that some of our analysis builds upon her work. We detail how our paper extends this concurrent work critiquing FLOP thresholds in the global response.
>
> **From useful documentation to meaningful regulations**
>
> Second, we appreciate your insightful comments on our discussion of data-centrism in AI governance. While many of the works we cite (including Datasheets and Data Nutritional Labels) undoubtedly advance dataset documentation, we argue that these existing approaches fall short in addressing the unique risks posed by aggregation and derivation in modern machine learning systems. These capabilities allow models to generate outcomes that exceed the apparent risks of individual data points or datasets in isolation. While we briefly touch on this in the current draft, we recognize that it may not have been adequately highlighted or organized.
>
> In our global response, we address how we will use dataset documentation methods effectively to address derivation and retrieval risks associated with the use of AI.
>
> **Gap between academia and regulation**
>
> Finally, we wish to emphasize the critical gap between academic progress in dataset documentation and its integration into regulatory frameworks. While academic efforts have made significant strides in standardizing documentation methods, these have not yet permeated regulatory discussions, as evidenced by our breakdown of the lack of data concerns in current governance documents. Our hope is to bridge this gap by encouraging not only policymakers but also the AI research community to integrate data-centric considerations more fully into their methods and discussions. We believe this dual approach—academic and regulatory—can yield meaningful improvements in how AI capabilities and risks are governed.
>
> ---
>
> We thank you for your detailed feedback and valuable suggestions. We look forward to improving the paper from this review and from further discussion in the remaining time.

---

> ### Comment · Reviewer_SyiS · 2024-11-26
>
> Thanks to the authors on the clarification of the contribution. Most impactful is their statement that this work is about how we can operationalise data governance research. However, I do not believe the author have provided with any frameworks to do so. I find their discussions not very different from other academic position papers - and unfortunately most of the arguments made in the paper have been made by other researchers. In this case, I will leave my score as is. If the authors were able to actually develop the regulation framework then I would regard this as a solid contribution and would vote to accept the paper. For now, I believe it might be worthwhile to have this paper accepted so that these discussions are brought to the fore at the conference but I am also okay if this paper does not go through because there is a lot of missing work.

---

> > ### Author Response · Authors · 2024-11-27
> >
> > Thank you for your continued engagement with our work and for providing detailed feedback. We appreciate your reflections and take them as an opportunity to clarify our contributions and goals.
> >
> > **Broader Scope from Prior Work** A key strength of our paper is the applicability beyond large language models (LLMs). While much of the discourse in AI governance has focused on LLMs, our work emphasizes trends that extend across vision models as well. Particularly, we are the first to demonstrate trends counter to the current AI governance paradigm not just in FLOPs like prior work, but also in parameter scaling. This is a critical shift, as it enables the discussion to remain relevant as AI technologies evolve. We aim to future-proof the conversation and avoid overly narrow definitions that may become obsolete.
> >
> > **Operationalizing Data Governance Research** The development of such a framework requires collective effort from the AI community. Rather than prescribing a fixed solution, our goal is to bring attention to critical gaps in existing discussions---specifically the challenges of derivation and retrieval risks.
> >
> > ---
> >
> > We have the shared belief that existing regulatory frameworks are limited and that the ICLR community is the right community to address these issues. Further, we agree that this work presents new analyses which extent existing work and challenge contemporary regulatory paradigms We thank you for recognizing the potential impact of having these discussions at ICLR and for your thoughtful consideration.

---

### Author Response · Authors · 2024-11-19
**Global response to all reviewers**

We thank all reviewers for their insightful questions and feedback. We are glad they found the paper clear and well-written [SyiS, SLu2], addressing a timely and important topic [SyiS, SLu2, ELVC], and the potential for timely impact [SyiS, ELVC].

We respond to concerns shared across reviewers below, and address individual comments in reviewer-specific responses.

**Relation to prior works on FLOP thresholds**

We thank the reviewers for highlighting previous work on FLOP thresholds (in particular, the work of Hooker 2024) and the broader efforts to emphasize the importance of dataset auditing and documentation.

We first want to clarify that our paper aims to extend Hooker’s contributions rather than replicate them. While her paper focuses specifically on the limitations of FLOPs thresholds in AI regulation and proposes dynamic thresholds as a solution, our work broadens the discussion to include both FLOPs and parameter counts. Additionally, her analysis primarily targets large-scale language models, whereas we demonstrate that similar limitations exist across a broader range of models, including smaller-scale vision models.

We will explicitly address this distinction in Section 2 and properly cite Hooker 2024 in our revisions.

**Further clarifications regarding governance proposals**

We greatly appreciate that the reviewers find value in this work and the topic we tackle. We hope that our paper continues to spark fruitful discourse in the ongoing discussion period, and later throughout the wider ICLR community.

Our proposal on operationalizing data-centric governance was not as clear as it could have been. Following, we will clarify our proposal in Section 5.2 on the use of a standardized ontology of “harmful” data. By applying this ontology to individual data points within documented datasets, we can generate aggregated risk profiles for models. For example, a dataset with 3,000 labeled data points related to “manipulation” could predict a model’s ability to perform downstream tasks such as political manipulation far more reliably than a dataset with 30 such points. Automating this process, leveraging documented datasets, could provide a scalable, first-pass risk assessment tool for derivation and retrieval.

Regarding governance, we have considered which agencies might be responsible for maintaining such risk ontologies and validating thresholds established through this process. However, given the fluid nature of governmental structures and the ongoing evolution of agency roles in AI governance, we intentionally avoid being overly prescriptive in the paper. Instead, our focus is on presenting a flexible framework that can be adapted to various governance contexts as these structures mature. Our aim is that some of the critical questions asked in this review - e.g., about implementation of data audits - are brought to the forefront of policymakers’ and ML researchers’ minds through this paper, so that these challenges can be addressed effectively.

---

Thank you for your reviews. They have already made our paper stronger. We are looking forward to discussing with you further.

---

### Meta-Review · Area_Chair_ft1D · 2024-12-16

**Metareview:**

After reading the reviewers' comments, and reviewing the paper, we regret to recommend rejection.

The paper is a perspective on AI governance, and highlights some of the key limitations in model-focused policies (that do not consider data). I believe the paper can be interesting to the AI community and spark discussions on an important topic.

However, there is no novel technical contribution, nor a clear framework to address the issue presented, but some recommendations.

**Additional Comments On Reviewer Discussion:**

The authors have been proactive in addressing the comments raised by the reviewers, and the reviewers were well engaged responding to the authors.

While taking into account the reviewers comments, and responses, we lean toward rejection, as we believe that the novelty of the paper may be lacking, and a clear framework for addressing the issue raised seems not clearly articulated.

No ethics review raised by the reviewers, and we agree with them.

---

### Decision · Program_Chairs · 2025-01-22

Reject